# Let's Think Var-by-Var: Large Language Models Enable *Ad Hoc* Probabilistic Reasoning

## Abstract

A hallmark of intelligence is the ability to flesh out underspecified situations using "common sense." We propose to extract that common sense from large language models (LLMs), in a form that can feed into probabilistic inference. We focus our investigation on *guesstimation* questions such as "How much are Airbnb listings in Newark, NJ?" Formulating a sensible answer without access to data requires drawing on, and integrating, bits of common knowledge about how `Price` and `Location` may relate to other variables, such as `Property Type`. Our framework answers such a question by synthesizing an *ad hoc* probabilistic model. First we prompt an LLM to propose a set of random variables relevant to the question, followed by moment constraints on their joint distribution. We then optimize the joint distribution $p$ within a log-linear family to maximize the overall constraint satisfaction. Our experiments show that LLMs can successfully be prompted to propose reasonable variables, and while the proposed numerical constraints can be noisy, jointly optimizing for their satisfaction reconciles them. When evaluated on probabilistic questions derived from three real-world tabular datasets, we find that our framework performs comparably to a direct prompting baseline in terms of total variation distance from the dataset distribution, and is similarly robust to noise.

## 1 Introduction

> Thus, in reasoning we depend very much on *prior information* to help us in evaluating the degree of plausibility in a new problem. This reasoning process goes on unconsciously, almost instantaneously, and we conceal how complicated it really is by calling it common sense. —E. T. Jaynes, *Probability Theory: The Logic of Science* (2003)

Humans constantly reason about novel situations, integrating evidence with prior knowledge. The Jaynes (2003) quote above refers to an everyday example: a policeman sees a masked man with a bag crawling out of the broken window of a jewelry store, and suspects a burglary. How can such conclusions be arrived at—appropriately generating hypotheses and weighing competing evidence?

Like Jaynes, we hope to draw on the very same methods of statistical modeling and inference that allow scientists to reason formally about complex domains like epidemiology, diplomacy, or syntax. For those domains, however, scientists normally invest time in perfecting a durable scientific model that supports many queries. Commonsense reasoning may instead generate a quick-and-dirty *ad hoc* model for each query.

We show that one can construct such ephemeral models automatically by enlisting the existing commonsense knowledge of large language models (LLMs). Of course, today's LLMs are already smart enough to recognize the above scene as a burglary—either at once, or via a chain of thought that may explicitly generate and evaluate different hypotheses. But there are harder situations that may benefit from systematically eliciting many fragments of relevant knowledge from the LLM, and deriving conclusions from this combined knowledge in a more formal and systematic way.

While one could elicit *logical* propositions and derive conclusions from those (Jung et al., 2022), we consider here the more general case of *probabilistic* knowledge and conclusions. Consider a

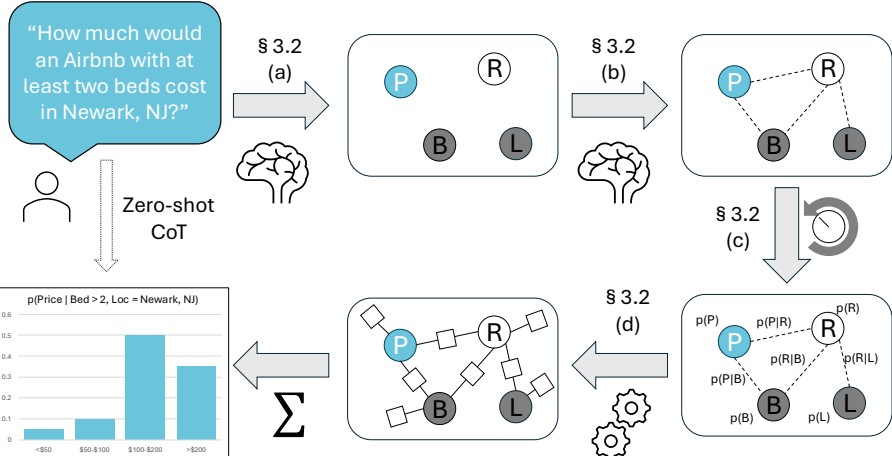

Figure 1: An illustration of our proposed framework applied to answering an example probabilistic question, $Q$ = "How much would an Airbnb with at least two beds cost in Newark, NJ?". Going **clockwise** from $Q$, we first prompt an LLM to brainstorm the relevant random variables (§3.2 (a)), producing Price (P), Rating (R), Beds (B), Location (L), where shaded nodes denote variables being conditioned on, blue nodes denote target variables, and white nodes denote latent variables. Then we prompt an LLM to propose interacting pairs $\{v_1, v_2\}$ of proposed variables, and whether to constrain $p(v_1 \mid v_2)$ or $p(v_2 \mid v_1)$ (§3.2 (b)). Next we prompt LLMs to propose numeric constraints on the marginal $p(v)$ of every proposed variable, as well as the conditional marginals $p(v_1 \mid v_2)$ of every proposed pairwise interaction (§3.2 (c)); Finally, we optimize the parameters of a log-linear model with fuzzy maximum entropy objective (2) in order to maximize constraint satisfaction (§3.2 (c)). The final output is an ad hoc probability model that can be used to answer $Q$. Going **counterclockwise** from $Q$ is a baseline of asking for an estimate of $Q$ directly using a zero-shot LLM with Chain-of-Thought.

*guesstimation* question such as "How many people in Nigeria own laptops?"[1] An LLM that has only weak intuitions about this may nonetheless be able to recall various relevant information:

- One route to an answer would estimate Nigeria's distribution over occupations, and then estimate those occupations' distributions over computing devices. It is relevant that Nigeria is a developing country and that some developing countries have largely skipped over laptops to mobile phones.

- Another route would estimate Nigeria's wealth distribution and its ownership rates for other appliances (cars, dishwashers, cellphones), and then guess how a person's laptop ownership correlates with their wealth and possessions.

- Another route might look at historical data (if known) and try to extrapolate to the present.

- The above bullets estimate Nigeria's *rate* of laptop ownership, which must be multiplied by Nigeria's population. If the population is not known, it could be guessed based other facts, such as Nigeria's physical size and political influence relative to nearby countries, or the relative visibility of Nigerians in global culture.

Integrating all of this information *systematically* may provide a more robust answer than simply asking the LLM to answer directly or to think step-by-step. We do this by constructing an *ad hoc* probability model over situations, with latent variables and their interactions proposed by the LLM.

Though the LLM proposes the model's structure, we do not expect the LLM to provide its *parameters*. In general, such parameters are not interpretable.[2] Rather, we ask the LLM to make predictions

---

[1]This may arise in the course of solving another guesstimation question: "If my aging laptop fails during my trip to Lagos, how long will it take to repair?"

[2]In a Markov random field (MRF), the optimal parameters for one factor are not a property of that factor alone, but depend strongly on what other factors have been added to the model and what their parameters are.

about the world—such as marginal probabilities. We set the model parameters so as to align the model's predictions with the LLM's predictions. We can then query our model to answer the original guesstimation question (via probabilistic inference over the situations described by the model variables).

This paper will focus on specific guesstimation questions where we (as experimenters) are able to evaluate answer quality. In §5, we evaluate our approach on three real-world datasets, Inside Airbnb (AIR)[3], American Time-Use Survey (ATUS)[4], and World Values Survey (WVS)[5], by comparing our system's answers to the answers estimated from these datasets. We develop our prompts on subsets of Inside Airbnb and American Time-Use Survey, and evaluate on held-out subsets of these two datasets, as well as on World Value Survey, which we held out completely during system development.

## 2 PROBLEM SETUP

Let $Q$ denote a question about some *novel* situation to some agent—in the sense that there is not enough prior experience to answer the question *directly*. Concretely, consider the example question, "What would the age be for a widow living in California?" Without direct prior knowledge (e.g. from having met many `Widows` in `California` or from looking up census data), formulating sensible answers to such questions requires drawing on and integrating bits of common knowledge about how `Widowness` and `Location` may relate to *other* variables like `Occupation` of their spouse, and whether they have any `Children`.

We can formalize such a question as a probabilistic query for a particular conditional distribution, $p(\mathrm{y} \mid \mathbf{x} \in \mathbb{S})$, where y is the target variable, $\mathbf{x}$ are the conditioning variables, and $\mathbb{S} \subseteq \mathbb{X}$ is the event being conditioned on. The example question above can be formalized this way as a query for $p(\texttt{Age} \mid \texttt{Location} = \texttt{California}, \texttt{Widow} = \texttt{True})$. Given such a question $Q$, our task is to generate an estimate $\hat{p}(\mathrm{y} \mid \mathbf{x} \in \mathbb{S})$ without relying on direct data.

For simplicity, our investigation will focus on questions where there is a single target variable, where all variables are discrete (we discretize continuous variables into ranges), but it would not be difficult to generalize our method to more than one target variable and to handle continuous variables directly (via approximate inference methods such as Minka (2013)).

## 3 METHOD

Our framework extracts knowledge from LLMs and integrates it to build an *ad hoc* probability model that can be used to answer the kind of questions described in §2. In §3.1, we formalize the notion of common knowledge relevant to some question $Q$, and how such knowledge can be integrated in a principled way to yield an *ad hoc* probability model. In §3.2, we describe how we instantiate the formalization with a prompted-LLM as the source of common knowledge, as well as specific choices we made in terms of parameterizing the ad hoc models.

### 3.1 INTEGRATING COMMON KNOWLEDGE VIA FUZZY MOMENT MATCHING

**Moment Constraints**     Given a question $Q$, for example, "What would the age be for a widow living in California?", what kinds of prior knowledge might be helpful for answering it? Our main insight is to extract prior knowledge in the form of *moment-matching constraints*, that is, constraints on the (conditional) marginals over random variables that are relevant to the question $Q$.

Let's suppose for now that we are supplied with a set of variables that are relevant to the question $Q$,[6] which includes the target variable y, the conditioning variables $\mathbf{x}$, and some latent variables $\mathbf{z}$. Our

---

[3] https://insideairbnb.com/

[4] https://www.bls.gov/tus/

[5] https://www.worldvaluessurvey.org/wvs.jsp

[6] The judgement of relevance of a random variable $\mathrm{x}_1$ to $\mathrm{x}_2$ is a kind of prior knowledge about their joint distribution. For example, relevance could be formalized as a threshold on the mutual information $I(\mathrm{x}_1, \mathrm{x}_2)$, which can be derived from their joint marginal.

constraints $c_1(p), \ldots c_n(p)$ on the conditional expectations of the joint distribution $p$ take the form

$$c_i(p) : b_i = \mathbb{E}_p \left[ f_i(\mathbf{y}, \mathbf{x}, \mathbf{z}) \mid g_i(\mathbf{y}, \mathbf{x}, \mathbf{z}) \right] \stackrel{\text{def}}{=} \frac{\mathbb{E}_{(\mathbf{y}, \mathbf{x}, \mathbf{z}) \sim p} \left[ f_i(\mathbf{y}, \mathbf{x}, \mathbf{z}) \cdot g_i(\mathbf{y}, \mathbf{x}, \mathbf{z}) \right]}{\mathbb{E}_{(\mathbf{y}, \mathbf{x}, \mathbf{z}) \sim p} \left[ \phantom{f_i(\mathbf{y}, \mathbf{x}, \mathbf{z}) \cdot} g_i(\mathbf{y}, \mathbf{x}, \mathbf{z}) \right]} \tag{1}$$

where $g_i$ is an indicator function and $f_i$ is a real-valued feature function.[7]

Why do we formalize prior knowledge as constraints on the *distribution* $p$ rather than its *parameters*? The optimal parameters of a probability distribution are often interdependent and change with the model structure. Adding new latent variables $\mathbf{z}$ to a model may change the optimal parameters in other parts of the model. However, conditional expectations are stable across different model structures since they are properties of the world, not properties of the model. This makes it possible to elicit them individually from an LLM.

**Estimation Objective**   The constraints will be drawn from an LLM and may not be wholly correct. We optimize $p$ to *approximately* satisfy the constraints via

$$\underset{p}{\text{argmin}} \, -H(p) + \sum_i w_i \left( b_i - \mathbb{E}_p \left[ f_i(\mathbf{y}, \mathbf{x}, \mathbf{z}) \mid g_i(\mathbf{y}, \mathbf{x}, \mathbf{z}) \right] \right)^2 \tag{2}$$

The hyperparameter $w_i$ specifies the importance of each constraint $c_i$, which controls tradeoffs when it is not possible to satisfy all constraints at once. Rewarding the Shannon entropy $H(p)$ encourages smoother distributions when it *is* possible to satisfy all constraints (Jaynes, 1957) and even when it is not. The hybrid objective (2) is historically known as the fuzzy maximum-entropy objective (Chen & Rosenfeld, 2000; Dudík et al., 2007) because it does not require the constraints to be satisfied exactly. Other reasonable variants are reviewed by Kazama & Tsujii (2005) and could be used here. Our innovation is to obtain the constraints from an LLM instead of from a data sample as in past work.

### 3.2 EXTRACTING COMMONSENSE FROM LLMs FOR PROBABILISTIC INFERENCE

We develop a concrete pipeline to build models as in §3.1 with LLMs as the knowledge source. In particular, the pipeline involves three stages of prompting: given a question $Q$, we identify (a) relevant variables and (b) pairs of interacting variables, allowing us to elicit (c) numerical constraints $\boldsymbol{c}$. We can then (d) formulate a log-linear family of distributions $p$ and optimize equation (2) over that family.

**(a) Brainstorming Relevant Variables**   Given a question $Q$ expressed in natural language, we prompt an LLM to brainstorm in free-form text, specifying the target variable y, the conditioning variables $\mathbf{x}$, and any additional variables $\mathbf{z}$ by giving them names as well as a list of possible values $\mathbb{Y}, \mathbb{X}, \mathbb{Z}$ that they can take on.

Specifically, we prompt with the system message in Appendix A.1.1 followed by the single (1-shot) example in Appendix A.1.2. The example's input is not from any of the domains we evaluate on; we obtained the example's output by lightly editing the 0-shot output from a strong LLM (namely GPT-4o).

We then prompt the LLM to translate this free-form answer into a machine-readable JSON object, including variable definitions.

For evaluation purposes, we also supply in user prompt name of the target variable y, all its possible values defined in the dataset, and encourage the LLM to include it in its variables. However, to ensure that the target variable is always used exactly, we do not extract it during translation and instead add it into the list of variables.

---

[7]In our experiments, $f_i$ will always be an indicator function as well, so the conditional expectations are simply conditional probabilities of our discrete random variables. However, allowing real-valued $f_i$ would let us constrain the means, variances, and covariances of random variables. In the future, we might further broaden the constraint language. For example, one might ask the LLM about the differences or ratios of conditional expectations—"cats weigh less than dogs on average"—or the conditional entropy or mutual information of random variables. The LLM could also be asked for prediction intervals rather than point estimates, resulting in interval constraints.

**(b) Choosing Quantities to Constrain**  We prompt the LLM to brainstorm interacting pairs of variables from stage (a), choose the best few pairs, and finally decide for each chosen pair $\{v_1, v_2\}$ whether to constrain $p(v_1 \mid v_2)$ or $p(v_2 \mid v_1)$. This prompt includes the brainstorming message from stage (a).

As before, we then prompt the LLM to translate this free-form list of conditional distributions into a JSON object. We then drop any $\mathbf{z}$ and $\mathbf{x}$ variables from the model that are not connected (directly or indirectly) to the target variable $\mathbf{y}$, and thus drop conditional distributions mentioning those variables.

**(c) Eliciting the Numerical Targets**  Now, for each surviving conditional distribution $p(v_1 \mid v_2)$, we ask the LLM to supply the numerical conditional probabilities. Specifically, for each $v_2 \in \mathbb{V}_2$, we prompt the LLM to generate a natural language query $Q'$ for the distribution $p(v_1 \mid v_2 = v_2)$, and then prompt the LLM separately to return that distribution as a vector of dimension $|\mathbb{V}_1|$.

(In principle, we could constrain the distribution of $v_1$ for only certain proposed values $v_2 = v_2$. We leave this possibility to future work, along the possibility of eliciting conditional or joint probabilities involving more than 2 variables.)

Using the same method of generating natural language questions $Q'$, we prompt for the unary marginal distribution $p(v)$ for each variable $v$. We similarly prompt for the distribution $p(\mathbf{y} \mid \mathbf{x} = \boldsymbol{x})$, which corresponds to the original question $Q$ (or a backed-off version of it, if some of the variables in $\mathbf{x}$ were dropped).

**(d) Optimizing a Log-linear Model**  We now choose a distribution $p$ that approximately has the elicited conditional and marginal probabilities, by optimizing equation (2). Specifically, we define a log-linear family of models $p_\theta$ and optimize $\theta$ by batch gradient descent. The features of the log-linear model are all and only the indicator functions $f_i$ and $g_i$ that are necessary to express the list of unary and pairwise constraints (but not necessarily $Q$). The factor graph of this joint model contains only pairwise and unary potential functions that correspond to the proposed constraints.

We use brute force summation to exactly compute the conditional probabilities in equation (2).[8] As for the weights $w_i$ in equation (2), we use $w_i = c$ for some constant $c$ to balance between constraint satisfaction and entropy smoothing.[9] We empirically choose $c$ on development data.

## 4 RELATED WORK

Large language models perform remarkably well on a diverse and challenging set of benchmarks (Ouyang et al., 2022; Anthropic Team, 2024; Gemini Team, 2024). Their effectiveness (Bubeck et al., 2023) is perhaps unsurprising, as they absorb vast amounts of world knowledge from their pretraining data (Petroni et al., 2019; AlKhamissi et al., 2022). On the other hand, their reasoning is brittle and is often based on shortcuts rather than sound inference rules (Saparov & He, 2023; **?**; Dziri et al., 2023). Some studies suggest that learning sound reasoning from samples may be too challenging due to statistical shortcuts (Geirhos et al., 2020), even if a deep architecture like Transformer (**?**) can in principle implement it (Zhang et al., 2022). Many methods have thus been developed to extract better reasoning from LLMs in hopes of making better predictions with them. Within this direction, two ideas are immediately relevant to our work.

The first idea is using LLMs to brainstorm various pieces of relevant common knowledge about a question and then aggregating them to arrive at a prediction. Wang et al. (2023); Yao et al. (2023); Besta et al. (2024); Jung et al. (2022) all do so by aggregating over multiple reasoning paths. Viewed through the lens of brainstorming relevant knowledge and aggregation, our work introduces a new unit of common knowledge—that of a moment constraint on a probability distribution. We also propose a corresponding aggregation procedure of optimizing a shared underlying probabilistic model to agree with all the constraints.

---

[8]In our experiments, we instruct the LLM to propose at most 4 variables, and to select no more edges than variables , which makes this feasible. Scaling up to larger models will require approximate inference algorithms which may introduce additional sources of error.

[9]A more sophisticated option would be to place more weight on constraints where the LLM is more confident in the target value $b_i$. Another possibility would be to downweight constraints on variables and pairs of variables with many values, so that the objective function is not dominated by the many constraints that they yield.

Another related idea is to augment LLMs with formal reasoning components such as external symbolic reasoning engines and soft verifiers (Lyu et al., 2023; Xu et al., 2024; Pan et al., 2023; Bostrom et al., 2022; Ling et al., 2023). Our method can be viewed as augmenting LLMs with a formal reasoning engine that includes both fuzzy moment matching to infer the parameters of a graphical model and probabilistic inference to make predictions from the graphical model. While the cited works focus on improving the *logical* reasoning of LLMs, we study how to improve the *probabilistic* reasoning of LLMs.

Particularly worth mentioning is the maieutic prompting method of Jung et al. (2022), which takes inspiration from both lines of ideas—they brainstorm latent propositions by abductive reasoning, and then solve a joint constraint satisfaction problem to guess which propositions are true (and in particular, whether the original query $Q$ is true). Their method can be viewed as performing *MAP* inference under a factor graph consisting of binary random variables corresponding to propositions, and with unary factors and binary factors whose parameters are extracted from LLMs and pretrained NLI models. They use a recursive algorithm to create an initial tree of propositions, and later add edges between all pairs of propositions. On the other hand, our method performs *marginal* inference over a factor graph of categorical variables corresponding to properties of situations in the world; our graph structure is directly proposed by an LLM is and usually sparser. The parameters of our graphical model are found by optimizing a set of LLM-proposed constraints on its various marginal distributions.

Probabilistic reasoning using LLMs has been relatively under-explored as a research problem. In a position paper, Dohan et al. (2022) propose to view prompted LLMs as conditional distributions over strings and the orchestration of LLM calls as a probabilistic program over strings (van de Meent et al., 2021). More recently, Nafar et al. (2024) use LLMs to generate probabilistic programs that get executed to produce distributions that answer probabilistic questions. However, crucially, their focus is more on abstract reasoning problems and requires as input the definition of a probabilistic model. Our work focuses on building that probabilistic model with the help of a LLM.

Researchers in Psychology and Cognitive Science have long explored the probability judgments in humans. Our work is also motivated by theories suggesting that a coherent probability judgment should be a accurate one. Osherson et al. (1994; 2001) proposed to extract from human intuitions a coherent distribution that reconciles a person's different instances of probability judgments. More recently, Zhu & Griffiths (2024) showed that LLMs exhibit similar statistical properties in their probability judgments. However, despite the theoretical soundness, empirical results in this area have been mixed (Zhu et al., 2022), and there often is a lack of correlation between a coherent judgement and an accurate judgment.

## 5 EXPERIMENTS

We perform two experiments. §5.1 studies whether our model-building pipeline helps end-to-end performance in answering questions of the form introduced in §2. §5.2 tests the effectiveness of our two prompting stages (§3.2), by measuring the effect of intervening on their results in various ways.[10] All of our experiments use the following setup.

**Task**    As described in §2, the task is to provide an estimate $\hat{p}$ (a normalized vector of size $|\mathbb{Y}|$) to a probability distribution $p(\mathbf{y} \mid \mathbf{x} \in \mathbb{S})$ described by a natural language question $Q$.

**Metric**    To evaluate the quality of an estimate $\hat{p}$, we compute its Total Variation Distance from a reference distribution $p$,

$$\text{TVD}(p, \hat{p}) = \frac{1}{2} \sum_{y \in \mathbb{Y}} |\hat{p}(y) - p(y)| \tag{3}$$

**Datasets**    To evaluate our system, we need questions $Q$ paired with reference distributions $p$. To do so, we derived questions from three publicly available tabular datasets spanning domains including short term rentals (Inside Airbnb), daily activities rental (American Time-Use Survey), and personal

---

[10]This may be reminiscent of interventional studies on internal activations of neural networks (mechanistic interpretability).

attitudes (World Values Survey). We first describe the datasets briefly, then how we generate a set of questions given the contents of the dataset.

The Inside Airbnb[11] dataset (**AIR**) is a publicly available dataset of property rental listings across cities in the United States during 2023. Data for a city is collected by Inside Airbnb if its part of a list of major cities, or upon community request. Among the available cities, we randomly sample six cities to use in our evaluation, plus one more for tuning prompts and hyper-parameters.

The American Time-Use Survey[12] (**ATUS**) is a publicly available census dataset that collects meta-data about how people in the United States spend their time over the course of the week. The data is published yearly, and we choose data from years 2018, 2020, 2022 for evaluation, while using 2023 data for development.

The World Values Survey[13] (**WVS**) is a survey dataset that collects demographic data about individuals in various countries and their responses to questions that probe their values. We randomly sample six countries for evaluation, and hold out this domain entirely for evaluation.

More details on the three datasets and their pre-processing is discussed in Appendix B.

**Question Generation**   We randomly sample formal probability queries with $n$ conditions based on the schema of the datasets, and translate them to natural language with the help of a LLM (we generate natural language questions given a formal query, and manually fix any errors). Specifically, for each dataset, and each $n \in \{0, 1, 2\}$, we first generate the set of all possible queries of the form $p(\text{y} \mid \text{x}_1 = x_1, \ldots, \text{x}_n = x_n)$, and then filter it down by requiring that at least one of the conditions changes the distribution over the target variable y by $\geq 0.05$ in terms of total variation distance. Then we sample 6 questions uniformly from this set. For comparability, the questions for a given dataset and $n$ are reused across all values of the split variable (city for AIR, year for ATUS, or country for WVS), with the question being additionally conditioned on this value. We refer to these as the `Main` questions.

For AIR and ATUS, we also generate a `Focus` set of questions by repeating the same sampling process described above, except with an additional filter that the target variable y must be `Price` or `Activity`, respectively. This provides a set of questions that is more focused. We chose `Activity` and `Price` because they potentially interact with many other random variables from their respective domains.

**LLM Calls**   Unless otherwise noted, we use GPT-4o-mini as the LLM in our experiments. All LLM calls are made at temperature 0.2, with a max token of 4096 (the default in LangChain OpenAI).

## 5.1   END-TO-END EVALUATION

We evaluate our pipeline end-to-end on the World Values Survey (WVS), which was not used to develop the pipeline. For completeness, we also evaluate on the held-out subsets of Inside Airbnb (AIR) and American Time-Use Survey (ATUS).

**Direct Prompting**   We compare against the obvious baseline of simply asking the LLM to answer $Q$, using a chain-of-thought prompt ("zero-shot CoT") at temperature 0.2. To ensure that the baseline enjoys a comparable amount of computation time, we actually call the LLM many times and average the resulting distributions $\hat{p}$. The number of calls is chosen to match the average number of calls made for extracting moment constraints in stage (b) of our pipeline.

**Restricted Variables**   We also report the performance of our pipeline when we prompt it to use only variables in the dataset's schema (see §5.2 below for details).

The results are given in Table 1. Figure 2 breaks them down by the number of conditions $|\mathbf{x}|$ specified in the question. Figure 4 in the appendices compares TVD of our method to the baseline on each question separately, using a scatterplot.

---

[11]https://insideairbnb.com/

[12]https://www.bls.gov/tus/

[13]https://www.worldvaluessurvey.org/wvs.jsp

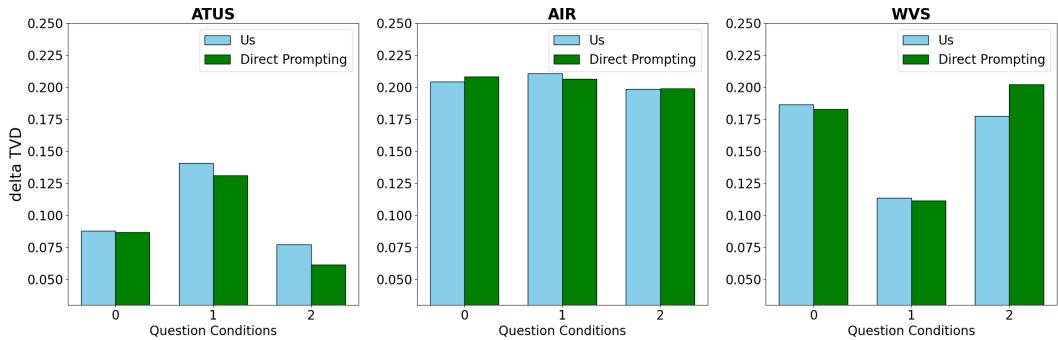

Figure 2: Breakdown of the end-to-end evaluation (§5.1) by number of conditions in the question.

Table 1: Average total variation distance against dataset distribution over questions as well as splits for subsets of questions `Main` and `Focus` respectively. We boldface the best result in each column along with all results that are not significantly worse (paired permutation test, $p < 0.05$).

|  | Main | | | Focus | | |
|---|---|---|---|---|---|---|
|  | ATUS | AIR | WVS | ATUS | AIR | WVS |
| Direct (§5.1) | **0.094** | 0.204 | **0.166** | **0.123** | **0.175** | — |
| Ours (§3) | **0.099** | **0.203** | **0.163** | **0.123** | 0.180 | — |
| Ours, restricted vars (§5.1) | **0.105** | **0.190** | — | 0.182 | **0.164** | — |

**Discussion of Results** Unfortunately, constructing and querying an *ad hoc* model was not more accurate than simply asking the LLM. The target questions $Q$ that we derived from these datasets were arguably too easy for our rather powerful LLM, GPT-4o-mini. The baseline system was already able to answer them with rather low TVD.

As a consolation, at least our method did not hurt. There are many ways that it could have gone wrong: after all, we were using natural language to obtain many imperfect numeric constraints and feeding them into a joint optimization problem. We had feared that the compounded noise in this process might swamp the signal. However, in practice the elicited constraints on both $Q$ and other conditional probabilities tended to be rather accurate in this domain.[14] Respecting these additional constraints simply did not change the answer much, either for better or for worse (see Figure 4).

Thus, an optimistic interpretation of the results is that our approach is viable, but that we would need to construct more difficult guesstimation problems or commonsense reasoning problems to show its value. Our approach will only help on problems where the LLM does not know how to answer the target question $Q$, but does know how to identify and answer other questions whose answers jointly imply an answer to $Q$.

We also discuss possible improvements to our method in §6, which might help on such a domain or on the current domain.

## 5.2 Intervention Experiments

We wish to study whether our method finds useful latent variables,[15] whether stage (b)'s proposed directions are helpful, and whether the elicited numeric constraints are accurate. This leads to the following set of interventions:

1. Randomly replacing a latent variable z with a different one after stages (a) and (b). This affects the natural-language questions that we ask at stage (c).

---

[14]We assessed them during pilot experiments on AIR to have an average TVD of 0.11. However, those results used the stronger GPT-4o model; we will add a formal evaluation using GPT-4o-mini.

[15]This requires stage (a) to propose the variables and also requires stage (b) not to discard them (see §3.2).

2. Randomly reverting the direction of the query $v_1 \mid v_2$ to be $v_2 \mid v_1$ after stage (b). Again, this affects the questions at stage (c).

3. Interpolating each elicited numeric constraint after stage (c) with the oracle value computed from the dataset.

For all intervention experiments, we omit the constraint on $p(y \mid \mathbf{x} = \boldsymbol{x})$, which corresponds to the original question $Q$. This constraint often has so much influence on the final result that it would mask the effect of the intervention.

Intervention 3 is possible only when the proposed variables appear as fields in the dataset so that we can get oracle values. Therefore, in that experiment—for both intervention 3 and its control condition—we modify the prompt of stage (a) to include the dataset schema (variable names along with their possible values) and to instruct the LLM to confine its brainstorming to these options.

We also use this modified prompt for intervention 1 and its control condition. This ensures a controlled comparison: it asks whether the LLM chooses wisely from among the schema variables, compared to the random choice of schema variable made by intervention 1. With the original prompt, the difference in performance might only reflect whether schema variables are more or less useful than non-schema variables.

1 and 2 are ablations that we expect to hurt performance. For 1, we randomly choose $i \in \{0, 1, 2\}$ number of variables that is not the target or the condition, and substitute uniformly from variables from the schema that's not already included. For 2, we randomly chose $j \in \{0, 1, 2, 3\}$ pairwise constraints to flip the direction. For both 1 and 2, since not all graphs have enough variables / edges that can be intervened on, we restrict our analysis to the subset of questions where the proposed model supports interventions of 3 node substitutions and 3 pairwise constraint reversals. (See Figure 3, columns 1 and 2.)

3 is an oracle intervention that we expect to help performance. We mix proposed distributions in stage (b) with the oracle distribution computed from the dataset at weight $w \in [0, 0.2, 0.4, 0.6, 0.8, 1.0]$, where $w = 0$ corresponds to no intervention, and $w = 1.0$ corresponds to using oracle numeric constraints. We also tried *hurting* performance by substituting a random distribution for the oracle distribution, drawing it uniformly from the simplex of probability distributions. (See Figure 3, columns 3 and 4.)

**Discussion of Results**   Columns 1 and 2 of Figure 3 suggest that perturbing the selection of variables or the direction of the conditional probabilities did not significantly affect the average gap between our method and the baseline. In other words, the LLM may not have made the best choices at these steps, despite our prompts.

Column 3 of Figure 3 provides a sanity check that as our constraints move towards the oracle, the error moves to 0. Unfortunately, this plot alone does not tease apart the contributions of moving the brainstorming queries produced by stages (a) and (b) towards oracle and moving the query corresponding to $Q$ towards the oracle. Even though we don't explicitly add it in the intervention experiments, stages (a) and (b) often propose a query corresponding to the question $Q$ by themselves. This suggests additional studies to separate the effect of a good answer $Q$ during brainstorming, and the effect of good answers to *other* related queries. Fortunately, column 4 of Figure 3 shows that artificial IID noise does not hurt our method by more than it hurts the direct-prompt baseline.

# 6   FUTURE WORK

Further prompt engineering might potentially help our system find crucial combinations of constraints that would improve on the baseline system. We cannot rule out the possibility that such constraints existed in our experiments and we simply failed to find them; we could use brute force exploration to check if they exist.

Stage (d) of our pipeline (§3.2) adds constraints to our model, but at the same time it expands the model family by creating additional parameters to help satisfy those constraints. As this may lead to overfitting, it might be wise to regularize our model objective (2) beyond the entropy term $H(p)$.

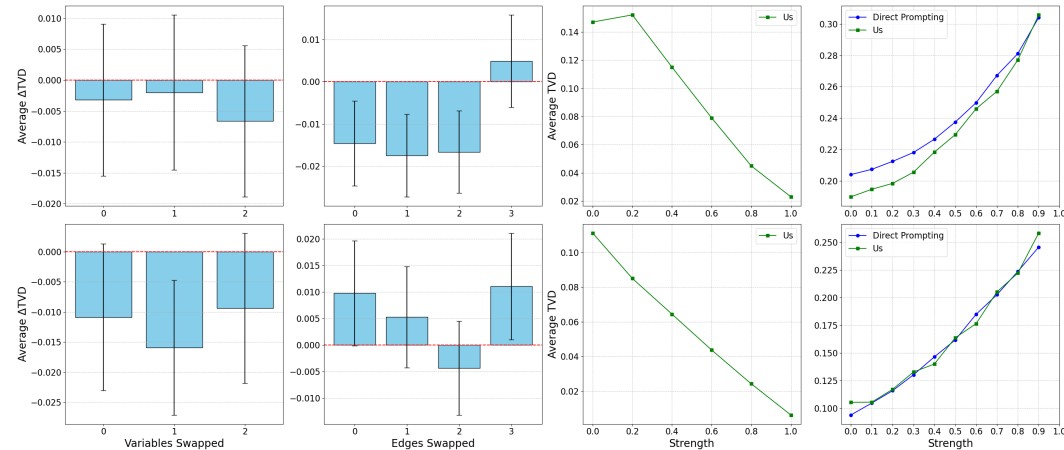

Figure 3: Results of intervention experiments (§5.1). "Us" in this figure refers to our approach. Top row corresponds to results on the `Main` set of questions on AIR domain, bottom row corresponds to the `Main` set of questions on ATUS domain. **Columns 1 and 2** visualize results of interventions 1 and 2, which randomly replaces zero to two latent variables with a different one after stages (a) and (b) of §3.2, and randomly reverses the direction of zero to three queries $v_1 \mid v_2$ to $v_2 \mid v_1$ after stage (b), respectively. Their $x$-axes denote the number of intervened nodes/queries, and their $y$-axes denote the average $\text{TVD}(p, \hat{p}_{\text{us}}) - \text{TVD}(p, \hat{p}_{\text{direct prompt}})$. The error bars denote one standard deviation of the average. Columns 3 and 4 correspond to intervention 3-oracle and intervention 3-noise. Their $x$-axes are the interpolation coefficient, and their $y$-axes are $TVD(p, \hat{p}_{\text{us}})$.

The LLM could also provide more precise information about how to penalize deviations from each constraint $c_i$, for example by providing a weight $w_i$, an interval on the target $b_i$, or a full loss function. The objective (2) could also be extended by asking the model $p$ to satisfy other kinds of constraints extracted from the LLM, such as relative probabilities (see footnote 7).

For simplicity, our implementation focused on models with a small number of categorical variables and only unary and binary factors. Future work should extend this to continuous variables as well as larger models, which may require approximate inference algorithms such as belief propagation and expectation propagation.

Our method builds an *ad hoc* model $p_\theta$ that can answer the original question $Q$, but $p_\theta$ can be interrogated further with additional probabilistic queries about its variables. Answers to those questions may be useful for interpreting the answer to the original question $Q$, and they may be compared against reference distributions computed from datasets to further assess the model.

Furthermore, $p_\theta$ can identify likely situations and marginally likely values for y and z. In principle, those could be fed back into a second round of brainstorming to further refine the model in high-probability regions of the outcome space—for example by introducing new latent variables or adjusting the granularity of existing variables.

We primarily used GPT-4o-mini for our experiments due to limited budgets. However, most LLM calls are spent on eliciting numerical targets in stage (c), we can use more powerful LLMs for stage (a) and (b), which can potentially improve the design of the ad hoc model.

Finally, future work should investigate when to trust the LLM. Confidence estimation could be used to upweight more accurate constraints in the optimization objective. In some cases, the LLM estimates might be improved (calibrated) with a small amount of supervised training data. For example, we might discover that the LLM tends to overestimate certain kinds of probabilities, and attempt to automatically correct these.

## ACKNOWLEDGMENTS

We thank X for the quote from Jaynes (2003), and for their helpful discussions.

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

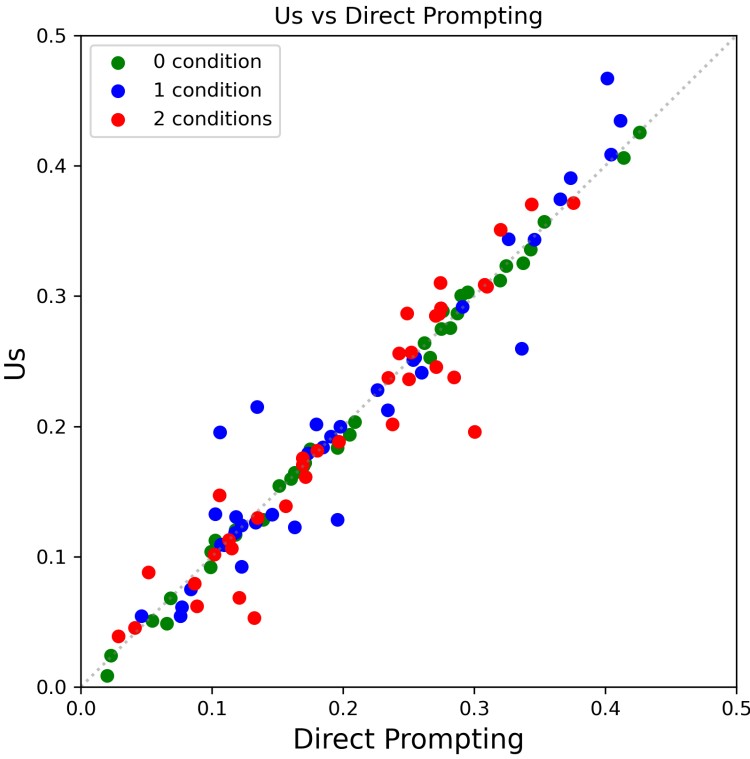

Figure 4: Scatterplot of the total variation distance against reference, Us versus Direct Prompt, on the `Main` set of questions for Inside Airbnb. Each point in the plot corresponds to a question from `Main` on a particular evaluation split (one of Ashville, Austin, Chicago, New Orleans, Pacific Grove, and Rhode Island), averaged over three random executions at temperature 0.2. The color of a point denote the number of conditions in the question. The other domains (ATUS and WVS) and the other set of questions (`Focus`) show a similar pattern in their scatterplots (not shown here).

Jian-Qiao Zhu and Thomas L. Griffiths. Incoherent probability judgments in large language models, 2024. URL `https://arxiv.org/abs/2401.16646`.

Jian-Qiao Zhu, Philip W.S. Newall, Joakim Sundh, Nick Chater, and Adam N. Sanborn. Clarifying the relationship between coherence and accuracy in probability judgments. *Cognition*, 223: 105022, 2022. ISSN 0010-0277. doi: https://doi.org/10.1016/j.cognition.2022.105022. URL `https://www.sciencedirect.com/science/article/pii/S0010027722000105`.

# A  APPENDIX

## A.1  PROMPTS

### A.1.1  VARIABLE PROPOSAL

SYSTEM

You are a data scientist.

You must design a graphical model to estimate conditional probabilities in a certain domain. The domain and the requested probabilities will be specified informally in REQUEST, so you must formalize the REQUEST into an Outcomes space, and defining categorical random Variables with mutually exclusive Values on the Outcomes space. The Outcomes space is a set of tuples that

represents all the possible Values combinations. DOMAIN, on the other hand, is one concise sentence that summarizes the entire Outcomes space. DOMAIN is thus a succinct summary of the population of the model, to provide to your colleagues. For simplicity, DOMAIN should omit information that the colleagues would assume by default. In other words, DOMAIN inlcudes simple background information, and only some Variables whose sets of Values on the Outcomes are non-trivial.

The Variables must be sufficient to answer the REQUEST and contain all possible Values on the Outcomes. That is, you must design your model so that the REQUEST can be formalized as $P(X_0 \mid X_1 \in x_1, \ldots, X_k \in x_k)$, where each $X_i$ is one of your model Variables and $x_i$ is some subset of its defined Values.

Although the REQUEST may specify Values of the Variables, the size of the model's Outcomes space is up to you to decide, i.e. you may define Values not specified in the REQUEST. This may allow us to use the model for other similar REQUESTs. In other words, the model may have an Outcomes space larger than implied in the REQUEST, on which the Variables can take Values that are not mentioned by REQUEST. If you choose a larger Outcomes space, make sure that the REQUEST can still be formalized exactly in the model.

You may also include additional Variables not mentioned by the REQUEST. These Variables can be very useful in mediating the relationships between $X_0$ and $X_1, \ldots X_k$. However, they should be concrete and unambiguous.

There can be at most 4 Variables in total, each with at most 5 Values. However, fewer Values enables more robust estimations. You can reduce the number of Values by grouping them into coarser categories, while still having enough granularity to express the REQUEST.

For simplicity, do not model missing data or edge cases, i.e. do not include Values like 'Unspecified' or 'Other'.

**Keep the Values categories concrete and precise. DO NOT USE Values like 'Other'.**

For Variables that are numeric in nature, use precise ranges.

Variables with one Value will not count toward the limit of 4, but still must be mentioned in the DOMAIN.

Some Variables may be supplied to you. Include them if appropriate.

Discuss the design details of your model. First identify the potential Variables in the REQUEST, and whether it will help to include other Values. Again, you are free to define Values not in the REQUEST so we can reuse the model, or fit the model with a larger dataset.

From this, decide the appropriate Outcomes size, i.e. whether a larger Outcomes will be helpful. Afterwards, define the Variables (including any additional Variables not in REQUEST) accordingly, with all possible Values on the Outcomes. Then, derive the DOMAIN from discussing each Variable's Values, and whether the Values set needs to be mentioned.

In the end, state the Variables, the DOMAIN in one sentence, and the formalized REQUEST.

### A.1.2 ONE-SHOT EXAMPLE FOR VARIABLE PROPOSAL

USER

REQUEST: For a horror movie released in United States between 1980 and 2000 rated as 'R', what is its probability distribution of total profit?

Supplied Variables:

Profit Range: 'Less than $10M'; '$10M - $50M'; '$50M - $100M'; 'Over $100M'

ASSISTANT

### Design Details

### Potential Variables in the REQUEST:

1. **Genre**: The genre is specified as horror.

2. **Country of Release**: This is specified as the United States.

3. **Release Period**: This is specified as between 1980 and 2000.

4. **Rating**: This is specified as 'R'.

5. **Profit Range**: Supplied with specific categories.

### Additional Variables:

1. **Production Budget**: This could be an important variable to include, as it can significantly affect the total profit.

2. **Critical Acclaim**: This is subjective and hard to obtain data for, so it will not be included.

### Deciding on the Outcomes Space

To make the model more reusable and provide more robust estimations, I will include other categories for the rating. This will allow us to use the model for other similar requests in the future. We also add production budget as an additional Variable.

### Variables and Their Values

1. **Genre**:

- Values: 'Horror' (only horror movies are of interest, fixed and does not count toward the limit of 4, mention in DOMAIN).

2. **Country of Release**:

- Values: 'United States' (only US releases are of interest, fixed and does not count toward the limit of 4, mention in DOMAIN).

3. **Release Period**:

- Values: '1980-1989'; '1990-2000' (to allow for some granularity in the release years, can be summarized as 1980 - 2000, mention in DOMAIN).

4. **Rating**:

- Values: 'G'; 'PG'; 'PG-13'; 'R'; 'NC-17' (enlarging the rating categories to make the model more reusable, summarized as all ratings, can be left out of DOMAIN).

5. **Profit Range**:

- Values: 'Less than $10M'; '$10M - $50M'; '$50M - $100M'; 'Over $100M' (as supplied, all profit ranges, can be left out of DOMAIN).

6. **Production Budget**:

- Values: 'Less than $10M'; '$10M - $50M'; 'Over $50M' (to provide granularity in the budget, summarized as all production ranges, can be left out of DOMAIN).

### DOMAIN

We need to mention the Horror genre, release in United States, released between 1980 - 2000. On the other hand, Rating (expanded), Profit Range, and Production Budget have covered all possible Values, and are rather trivial to mention. We summarize the model population as:

'Horror movies released in the United States between 1980 and 2000'

### Formalized REQUEST

The formalized REQUEST is: P(Profit Range | Genre = 'Horror', Country of Release = 'United States', Release Period ∈ {'1980-1989', '1990-2000'}, Rating = 'R').

### Summary

**Variables:**

1. **Genre**:

- Values: 'Horror'

2. **Country of Release**:

- Values: 'United States'

3. **Release Period**:

- Values: '1980-1989'; '1990-2000'

4. **Rating**:

- Values: 'G'; 'PG'; 'PG-13'; 'R'; 'NC-17'

5. **Profit Range**:

- Values: 'Less than $10M'; '$10M - $50M'; '$50M - $100M'; 'Over $100M'

6. **Production Budget**:

- Values: 'Less than $10M'; '$10M - $50M'; 'Over $50M'

**DOMAIN**: 'Horror movies released in the United States between 1980 and 2000.'

**Formalized REQUEST**: P(Profit Range | Genre = 'Horror', Country of Release = 'United States', Release Period $\in$ {'1980-1989', '1990-2000'}, Rating = 'R')

### A.1.3 VARIABLE PROPOSAL EXAMPLE DIALOGUE

USER

REQUEST: For an Airbnb listing in Austin, TX in 2023 with a price between $201 to $500 and is listed to accommodate a maximum of 4 or more people, what is the probability distribution of room type?

Supplied Variables:

Room Type: 'entire home or apartment'; 'private room'; 'shared or hotel room'

ASSISTANT

### Design Details

#### Potential Variables in the REQUEST:

1. **Location**: This is specified as Austin, TX.

2. **Year**: This is specified as 2023.

3. **Price Range**: This is specified as between $201 and $500.

4. **Maximum Occupancy**: This is specified as accommodating 4 or more people.

5. **Room Type**: This is supplied with specific categories.

#### Additional Variables:

1. **Property Type**: This could be useful to include as it may influence the room type (e.g., house, apartment, condo).

2. **Amenities**: This could also be relevant, but it is too broad and subjective for this model.

### Deciding on the Outcomes Space

To make the model more reusable, I will include additional categories for the property type. This will allow us to use the model for other similar requests in the future. The price range and maximum occupancy will also be defined clearly.

### Variables and Their Values

1. **Location**:

- Values: 'Austin, TX' (fix to Austin, does not count toward the limit of 4, mention in DOMAIN).

2. **Year**:

- Values: '2023' (fix to 2023, does not count toward the limit of 4, mention in DOMAIN).

3. **Price Range**:

- Values: '$201 - $300'; '$301 - $400'; '$401 - $500' (providing granularity within the specified range).

4. **Maximum Occupancy**:

- Values: '1-3 people'; '4 or more people' (to capture the specified condition).

5. **Room Type**:

- Values: 'Entire home or apartment'; 'Private room'; 'Shared or hotel room' (as supplied).

6. **Property Type**:

- Values: 'House'; 'Apartment'; 'Condo'; 'Townhouse'; 'Other' (to provide additional context for the listing).

### DOMAIN

We need to mention the location (Austin, TX), the year (2023), and the price range (between $201 and $500). The maximum occupancy condition is also relevant but can be summarized in the context of the room type. The room type and property type have covered all possible values and are rather trivial to mention. We summarize the model population as:

"Airbnb listings in Austin, TX in 2023 with a price between $201 and $500."

### Formalized REQUEST

The formalized REQUEST is: P(Room Type | Location = 'Austin, TX', Year = '2023', Price Range $\in$ {'$201 - $300', '$301 - $400', '$401 - $500'}, Maximum Occupancy = '4 or more people').

### Summary

**Variables:**

1. **Location**:

- Values: 'Austin, TX'

2. **Year**:

- Values: '2023'

3. **Price Range**:

- Values: '$201 - $300'; '$301 - $400'; '$401 - $500'

4. **Maximum Occupancy**:

- Values: '1-3 people'; '4 or more people'

5. **Room Type**:

- Values: 'Entire home or apartment'; 'Private room'; 'Shared or hotel room'

6. **Property Type**:

- Values: 'House'; 'Apartment'; 'Condo'; 'Townhouse'; 'Other'

**DOMAIN**: "Airbnb listings in Austin, TX in 2023 with a price between $201 and $500."

**Formalized REQUEST**: P(Room Type | Location = 'Austin, TX', Year = '2023', Price Range $\in$ {'$201 - $300', '$301 - $400', '$401 - $500'}, Maximum Occupancy = '4 or more people').

A.1.4  VARIABLE PROPOSAL EXAMPLE TRANSLATION

SYSTEM You will be given the transcript of a statistician's model designing process. Transcribe the final Variables and DOMAIN in the provided message into JSON using a provided schema. You can

find the Variables and DOMAIN toward the end of the message in a summary. Do not extract Room Type.

USER

### Design Details

#### Potential Variables in the REQUEST:

1. **Location**: This is specified as Austin, TX.

2. **Year**: This is specified as 2023.

3. **Price Range**: This is specified as between $201 and $500.

4. **Maximum Occupancy**: This is specified as accommodating 4 or more people.

5. **Room Type**: This is supplied with specific categories.

#### Additional Variables:

1. **Property Type**: This could be useful to include as it may influence the room type (e.g., house, apartment, condo).

2. **Amenities**: This could also be relevant, but it is too broad and subjective for this model.

### Deciding on the Outcomes Space

To make the model more reusable, I will include additional categories for the property type. This will allow us to use the model for other similar requests in the future. The price range and maximum occupancy will also be defined clearly.

### Variables and Their Values

1. **Location**:

- Values: 'Austin, TX' (fix to Austin, does not count toward the limit of 4, mention in DOMAIN).

2. **Year**:

- Values: '2023' (fix to 2023, does not count toward the limit of 4, mention in DOMAIN).

3. **Price Range**:

- Values: '$201 - $300'; '$301 - $400'; '$401 - $500' (providing granularity within the specified range).

4. **Maximum Occupancy**:

- Values: '1-3 people'; '4 or more people' (to capture the specified condition).

5. **Room Type**:

- Values: 'Entire home or apartment'; 'Private room'; 'Shared or hotel room' (as supplied).

6. **Property Type**:

- Values: 'House'; 'Apartment'; 'Condo'; 'Townhouse'; 'Other' (to provide additional context for the listing).

### DOMAIN

We need to mention the location (Austin, TX), the year (2023), and the price range (between $201 and $500). The maximum occupancy condition is also relevant but can be summarized in the context of the room type. The room type and property type have covered all possible values and are rather trivial to mention. We summarize the model population as:

"Airbnb listings in Austin, TX in 2023 with a price between $201 and $500."

### Formalized REQUEST

The formalized REQUEST is: P(Room Type | Location = 'Austin, TX', Year = '2023', Price Range $\in$ {'$201 - $300', '$301 - $400', '$401 - $500'}, Maximum Occupancy = '4 or more people').

### Summary

**Variables:**

1. **Location**:

- Values: 'Austin, TX'

2. **Year**:

- Values: '2023'

3. **Price Range**:

- Values: '$201 - $300'; '$301 - $400'; '$401 - $500'

4. **Maximum Occupancy**:

- Values: '1-3 people'; '4 or more people'

5. **Room Type**:

- Values: 'Entire home or apartment'; 'Private room'; 'Shared or hotel room'

6. **Property Type**:

- Values: 'House'; 'Apartment'; 'Condo'; 'Townhouse'; 'Other'

**DOMAIN**: "Airbnb listings in Austin, TX in 2023 with a price between $201 and $500."

**Formalized REQUEST**: P(Room Type | Location = 'Austin, TX', Year = '2023', Price Range $\in$ {'$201 - $300', '$301 - $400', '$401 - $500'}, Maximum Occupancy = '4 or more people').

ASSISTANT

{"Variables": [{"Name": "Location", "Value": ["Austin, TX"]}, {"Name": "Year", "Value": ["2023"]}, {"Name": "Price Range", "Value": ["$201 - $300", "$301 - $400", "$401 - $500"]}, {"Name": "Maximum Occupancy", "Value": ["1-3 people", "4 or more people"]}, {"Name": "Room Type", "Value": ["Entire home or apartment", "Private room", "Shared or hotel room"]}, {"Name": "Property Type", "Value": ["House", "Apartment", "Condo", "Townhouse", "Other"]}], "DOMAIN": "Airbnb listings in Austin, TX in 2023 with a price between $201 and $500."}

A.1.5 INTERACTION PROPOSAL EXAMPLE DIALOGUE

SYSTEM

You are designing an undirected graphical model for DOMAIN The model will consist of only binary factors. You will be given a list of categorical variable definition with Target variable and potentially some Observed variables and Latent variables. Propose pairs of variables from this list that should be coupled by a factor in your model.

Your model should be good at predicting Target VARIABLE when observing Observed variables. Latent variables may help reasoning about different situation and contexts.

You are free to omit Observed or Latent variables that are not helpful for this prediction. Some Latent variables may exist only to aid prediction of other Observed or Latent variables, and not directly correlate with the Target variable.

For a potential factor between two variables, consider their relationship IN STRICT ISOLATION, i.e. WITHOUT referring to any other variable. To do this, when considering (A, B), ALWAYS explicitly state a question: does A vary significantly with different values of B? Then answer the question. If you are not very confident, it means they should not be connected.

Avoid focusing on only the Target variable. Aim for few but strong factors. Add factors sparingly. Specifically, with N variables, NEVER propose more than N factors. For the variables you do include in the model, the variables should form a connected graph.

Start by repeating the maximum number of factors allowed, then discuss the model design.

After potential options, select a final set of few but strong factors. For each factor, consider whether you will be better at guessing the values of p(A|B) or p(B|A). If P(A|B) is easier to guess, list them in an ordered pair (A, B) to represent (Target, Condition). Otherwise, if P(B|A) is easier to guess, give (B, A).

In the end, state them a list, each of which is an ordered pair of 2 strings.

USER

**Target Variable:**

Room Type: 'entire home or apartment'; 'private room'; 'shared or hotel room'

**Observed Variable:**

Maximum Occupancy: '1-3 people'; '4 or more people'

**Latent Variable:**

Price Range: '$201 - $300'; '$301 - $400'; '$401 - $500'

Property Type: 'House'; 'Apartment'; 'Condo'; 'Townhouse'

ASSISTANT

The maximum number of factors allowed is equal to the number of variables in the model. In this case, we have 5 variables: 1 Target Variable, 1 Observed Variable, and 3 Latent Variables. Therefore, we can propose a maximum of 5 factors.

### Model Design Discussion

1. **Room Type (Target Variable)** and **Maximum Occupancy (Observed Variable)**:

- Question: Does Room Type vary significantly with different values of Maximum Occupancy?

- Answer: Yes, different room types can accommodate different maximum occupancies. For example, an entire home or apartment typically accommodates more people than a shared or hotel room.

2. **Room Type (Target Variable)** and **Price Range (Latent Variable)**:

- Question: Does Room Type vary significantly with different values of Price Range?

- Answer: Yes, different room types are often associated with different price ranges. For instance, entire homes or apartments may be priced higher than shared or hotel rooms.

3. **Room Type (Target Variable)** and **Property Type (Latent Variable)**:

- Question: Does Room Type vary significantly with different values of Property Type?

- Answer: Yes, the type of property can influence the room type. For example, a house is more likely to be listed as an entire home or apartment compared to a condo.

4. **Maximum Occupancy (Observed Variable)** and **Price Range (Latent Variable)**:

- Question: Does Maximum Occupancy vary significantly with different values of Price Range?

- Answer: Yes, properties that accommodate more people may also be priced differently, often higher.

5. **Maximum Occupancy (Observed Variable)** and **Property Type (Latent Variable)**:

- Question: Does Maximum Occupancy vary significantly with different values of Property Type?

- Answer: Yes, different property types can have different maximum occupancies. For example, a townhouse may accommodate more people than a condo.

### Final Set of Factors

After considering the relationships, I will select the following pairs of variables to form the factors in the model:

1. (Room Type, Maximum Occupancy) - It is easier to guess the Room Type given the Maximum Occupancy.

2. (Room Type, Price Range) - It is easier to guess the Room Type given the Price Range.

3. (Room Type, Property Type) - It is easier to guess the Room Type given the Property Type.

4. (Maximum Occupancy, Price Range) - It is easier to guess the Maximum Occupancy given the Price Range.

5. (Maximum Occupancy, Property Type) - It is easier to guess the Maximum Occupancy given the Property Type.

### List of Factors

1. (Maximum Occupancy, Room Type)

2. (Price Range, Room Type)

3. (Property Type, Room Type)

4. (Price Range, Maximum Occupancy)

5. (Property Type, Maximum Occupancy)

This selection of factors creates a connected graph that effectively captures the relationships between the variables while focusing on strong predictive capabilities for the target variable.

A.1.6   INTERACTION PROPOSAL EXAMPLE TRANSLATION

SYSTEM You will be provided a message transcript, where at the end there is a list of factor pairs in the form of [Target, Condition]. Translate the proposed factor pairs with Target and Condition into JSON. You must use the names exactly as provided at the very end of the message, under 'VARIABLE NAMES'

USER

The maximum number of factors allowed is equal to the number of variables in the model. In this case, we have 5 variables: 1 Target Variable, 1 Observed Variable, and 3 Latent Variables. Therefore, we can propose a maximum of 5 factors.

### Model Design Discussion

1. **Room Type (Target Variable)** and **Maximum Occupancy (Observed Variable)**:

- Question: Does Room Type vary significantly with different values of Maximum Occupancy?

- Answer: Yes, different room types can accommodate different maximum occupancies. For example, an entire home or apartment typically accommodates more people than a shared or hotel room.

2. **Room Type (Target Variable)** and **Price Range (Latent Variable)**:

- Question: Does Room Type vary significantly with different values of Price Range?

- Answer: Yes, different room types are often associated with different price ranges. For instance, entire homes or apartments may be priced higher than shared or hotel rooms.

3. **Room Type (Target Variable)** and **Property Type (Latent Variable)**:

- Question: Does Room Type vary significantly with different values of Property Type?

- Answer: Yes, the type of property can influence the room type. For example, a house is more likely to be listed as an entire home or apartment compared to a condo.

4. **Maximum Occupancy (Observed Variable)** and **Price Range (Latent Variable)**:

- Question: Does Maximum Occupancy vary significantly with different values of Price Range?

- Answer: Yes, properties that accommodate more people may also be priced differently, often higher.

5. **Maximum Occupancy (Observed Variable)** and **Property Type (Latent Variable)**:

- Question: Does Maximum Occupancy vary significantly with different values of Property Type?

- Answer: Yes, different property types can have different maximum occupancies. For example, a townhouse may accommodate more people than a condo.

### Final Set of Factors

After considering the relationships, I will select the following pairs of variables to form the factors in the model:

1. (Room Type, Maximum Occupancy) - It is easier to guess the Room Type given the Maximum Occupancy.

2. (Room Type, Price Range) - It is easier to guess the Room Type given the Price Range.

3. (Room Type, Property Type) - It is easier to guess the Room Type given the Property Type.

4. (Maximum Occupancy, Price Range) - It is easier to guess the Maximum Occupancy given the Price Range.

5. (Maximum Occupancy, Property Type) - It is easier to guess the Maximum Occupancy given the Property Type.

### List of Factors

1. (Maximum Occupancy, Room Type)

2. (Price Range, Room Type)

3. (Property Type, Room Type)

4. (Price Range, Maximum Occupancy)

5. (Property Type, Maximum Occupancy)

This selection of factors creates a connected graph that effectively captures the relationships between the variables while focusing on strong predictive capabilities for the target variable.

**VARIABLE NAMES:** (Use Name exactly as defined below)

'Price Range'; 'Maximum Occupancy'; 'Property Type'; 'Room Type';

ASSISTANT

{"Features": [{"Target": {"Name": "Room Type"}, "Condition": {"Name": "Maximum Occupancy"}}, {"Target": {"Name": "Room Type"}, "Condition": {"Name": "Price Range"}}, {"Target": {"Name": "Room Type"}, "Condition": {"Name": "Property Type"}}, {"Target": {"Name": "Maximum Occupancy"}, "Condition": {"Name": "Price Range"}}, {"Target": {"Name": "Maximum Occupancy"}, "Condition": {"Name": "Property Type"}}]}

### A.1.7 QUESTION GENERATION EXAMPLE

SYSTEM You will translate a formalized request for a (conditional) distribution into a concise question expressed in natural language. The formalized request contains three entries: DOMAIN, TARGET, CONDITION. formalized request is for P(TARGET | X_0 $\in$ x_0,... , X_k $\in$ x_k) under the DOMAIN.

DOMAIN provides background information for the request.

TARGET is the VARIABLE whose distribution is requested.

CONDITION provide a (possibly empty) list of conditions, each in the form 'X_i $\in$ x_i'. The request is conditioned on VARIABLE X_i taking any value in the set x_i.

USER

DOMAIN: Airbnb listings in Austin, TX in 2023 with a price between \$201 and \$500.

TARGET: Maximum Occupancy: ['1-3 people'; '4 or more people']

CONDITION: Price Range $\in$ 2208 ['\$401 - \$500']

ASSISTANT

For an Airbnb listing in Austin, TX in 2023 with a price between $201 to $500 and is listed to accommodate a maximum of 4 or more people, what is the probability distribution of room type?

USER

DOMAIN: Airbnb listings in Austin, TX in 2023 with a price between $201 and $500.

TARGET: Maximum Occupancy: ['1-3 people'; '4 or more people']

CONDITION: Price Range ∈ ['$401 - $500']

ASSISTANT

For an Airbnb listing in Austin, TX in 2023 priced between $401 and $500, what is the probability distribution of maximum occupancy?

### A.1.8 NUMERIC CONSTRAINT PROPOSAL EXAMPLE DIALOGUE

SYSTEM

You will estimate a probability distribution given the definition of the Target Variable. For each probability, give the best exact estimate instead of intervals.

USER

For an Airbnb listing in Austin, TX in 2023 priced between $401 and $500, what is the probability distribution of maximum occupancy?

'1-3 people'; '4 or more people'

ASSISTANT

To estimate the probability distribution of maximum occupancy for an Airbnb listing in Austin, TX, priced between $401 and $500 in 2023, we can consider general trends in the market.

Given the price range, it's likely that listings in this category are more spacious and may cater to larger groups. Therefore, the probability distribution can be estimated as follows:

- Probability of '1-3 people': 30%

- Probability of '4 or more people': 70%

This distribution reflects the expectation that higher-priced listings are more likely to accommodate larger groups.

### A.1.9 NUMERIC CONSTRAINT PROPOSAL EXAMPLE TRANSLATION

SYSTEM You will extract a statistician's probability distribution estimate from a given message into JSON. The estimate is for the Variable:

Target: Maximum Occupancy: '1-3 people'; '4 or more people'

Use exactly this naming and ordering of the Variable Value for the extraction. Extract a distribution of 2 numbers corresponding to each of its Value.

USER

To estimate the probability distribution of maximum occupancy for an Airbnb listing in Austin, TX, priced between $401 and $500 in 2023, we can consider general trends in the market.

Given the price range, it's likely that listings in this category are more spacious and may cater to larger groups. Therefore, the probability distribution can be estimated as follows:

- Probability of '1-3 people': 30%

- Probability of '4 or more people': 70%

This distribution reflects the expectation that higher-priced listings are more likely to accommodate larger groups.

ASSISTANT

{'Target': {'Name': 'Maximum Occupancy', 'Value': ['1-3 people', '4 or more people']}, 'Probability': [30, 70]}

PROCESSED

{"Target": {"Name": "Maximum Occupancy", "Value": ["1-3 people", "4 or more people"]}, "Condition": {"Name": "Price Range", "Value": ["$401-500$"]}, "Probability": [0.3, 0.7]}

### A.1.10 ZERO-SHOT CHAIN-OF-THOUGHT EXAMPLE DIALOGUE

SYSTEM

You will estimate a probability distribution given the definition of the Target Variable. For each probability, give the best exact estimate instead of intervals.

USER

For an Airbnb listing in Austin, TX in 2023 with a price between $201 to $500 and is listed to accommodate a maximum of 4 or more people, what is the probability distribution of room type? 'entire home or apartment'; 'private room'; 'shared or hotel room'

ASSISTANT

To estimate the probability distribution of room types for an Airbnb listing in Austin, TX in 2023, given the specified criteria (price between $201 to $500 and accommodating 4 or more people), we can provide the following estimates based on trends observed in the market:

- **Entire home or apartment**: 70%

- **Private room**: 25%

- **Shared or hotel room**: 5%

These estimates reflect the likelihood of each room type being listed under the given conditions.

### A.1.11 ZERO-SHOT CHAIN-OF-THOUGHT EXAMPLE TRANSLATION

SYSTEM

You will extract a statistician's probability distribution estimate from a given message into JSON. The estimate is for the Variable: Room Type: 'entire home or apartment'; 'private room'; 'shared or hotel room'

Use exactly this naming and ordering of the Variable Value for the extraction. Extract a distribution of 3 numbers corresponding to each of its Value.

USER

To estimate the probability distribution of maximum occupancy for an Airbnb listing in Austin, TX, priced between $401 and $500 in 2023, we can consider general trends in the market.

Given the price range, it's likely that listings in this category are more spacious and may cater to larger groups. Therefore, the probability distribution can be estimated as follows:

- Probability of '1-3 people': 30%

- Probability of '4 or more people': 70%

This distribution reflects the expectation that higher-priced listings are more likely to accommodate larger groups.

ASSISTANT

{"Target": {"Name": "Room Type", "Value": ["entire home or apartment", "private room", "shared or hotel room"]}, "Probability": [70, 25, 5]}

PROCESSED

{"Target": {"Name": "Room Type", "Value": ["entire home or apartment", "private room", "shared or hotel room"]}, "Probability": [0.7, 0.25, 0.05]}

## B  ADDITIONAL DETAILS ON DATASET AND PREPROCESSING

### B.1  DATASET SPLITS

1. On Inside Airbnb, we use Ashville, Austin, Chicago, New Orleans, Pacific Grove, and Rhode Island for evaluation, and Twin Cities for development.

2. On American Time-Use survey, we use 2018, 2020, 2022 as evaluation, and 2023 as development.

3. On World Values Survey, we use Malaysia, New Zealand, Rwanda, Sweden, United States, and Uruguay for evaluation, and no development.

### B.2  PREPROCESSING

For each dataset, we use a subset of all available columns. We also discretize any continuous data into ranges, and coarsen any discrete variables with too many values. All such choices were made before any significant tuning of the prompts and hyper-parameters of our pipeline or the prompt for zero-shot Chain-of-Thought baseline.

#### B.2.1  INSIDE AIRBNB

Many columns of the Inside Airbnb dataset have missing values for a significant proportion of rows. We thus ignored any column with too high a proportion of missing values, and then manually picked a subset of 8 columns that we judged to be interesting. The processed variables and their possible values are included in Table 2.

Table 2: Schema for our processed Inside Airbnb dataset.

| Column Name | Possible Values |
|---|---|
| Number of Bedrooms | studio or 1 bedroom, 2 bedrooms, 3 bedrooms, 4 or more bedrooms |
| Number of Bathrooms | shared or single bathroom, 2 bathrooms, 3 or more bathrooms |
| Superhost Status | Superhost, Not Superhost |
| Room Type | entire home or apartment, private room, shared or hotel room |
| Total Beds | 1 bed, 2 beds, 3 beds, 4 or more beds |
| Review Score | less than 4.4, 4.5 to 4.8, at least 4.9 |
| Max Accommodates | 1, 2, 3, 4 or more |
| Price | under $50, $51 to $100, $101 to $200, $201 to $500, at least $501 |

#### B.2.2  AMERICAN TIME-USE SURVEY

We use most of the frequently used subset[16] of ATUS. The processed variables and their possible values are included in Table 3.

#### B.2.3  WORLD VALUES SURVEY

Again, we manually picked most of the objective demographics variables as well as columns that are not too granular. The processed variables and their possible values are included in Table 4.

## C  EXAMPLE QUESTIONS

### C.1  AIRBNB

**Split**: Chicago, IL

---

[16]https://www.bls.gov/tus/other-documentation/freqvariables.pdf

Table 3: Schema for our processed American Time-Use Survey dataset.

| Column Name | Possible Values |
|---|---|
| Sex | Male, Female |
| Age | 15-29, 30-44, 45-64, 65-85 |
| Region | Northeast, Midwest, South, West |
| Marital Status | Married, Widowed, Divorced, Separated, Never Married |
| Metropolitan Residency Status | Metropolitan, Non-metropolitan |
| Labor Force Status | Employed, Unemployed, Not in Labor Force |
| Household Composition | Children Under 18 Present in Household, No Children Under 18 in Household |
| Day of Week | Weekday, Weekend |
| High School/College Enrollment | Currently Enrolled, Not Currently Enrolled |
| Activity | Personal Care, Sleep, and Sustenance, Leisure, Sports, and Social, Traveling and Commuting, Work and Education, Household and Other |

Table 4: Schema for our processed World Values Survey dataset.

| Column Name | Possible Values |
|---|---|
| Importance of family in life | Not at all important, Not very important, Rather important, Very important |
| Importance of friends in life | Not at all important, Not very important, Rather important, Very important |
| Importance of leisure time in life | Not at all important, Not very important, Rather important, Very important |
| Importance of politics in life | Not at all important, Not very important, Rather important, Very important |
| Importance of work in life | Not at all important, Not very important, Rather important, Very important |
| Importance of religion in life | Not at all important, Not very important, Rather important, Very important |
| Member of religious organization | Member, Not member |
| Member of sport or recreational organization | Member, Not member |
| Member of art, music or educational organization | Member, Not member |
| Member of labour union | Member, Not member |
| Member of political party | Member, Not member |
| Member of environmental organization | Member, Not member |
| Member of humanitarian or charitable organization | Member, Not member |
| Marital Status | Married, Divorced, Separated, Widowed, Single |
| Age | 18-29, 30-44, 45-64, 65+, |
| Sex | Male, Female |
| Labor Force Status | Employed, Unemployed, Not in Labor Force |

**Target**:
Number of Bathrooms: shared or single bathroom; 2 bathrooms; 3 or more bathrooms

**Conditions**:
Number of Bedrooms = 3 Bedrooms

**Natural Language Question**:
For an Airbnb listing with 3 bedrooms in Chicago, IL in 2023, what is the probability distribution of its number of bathrooms?

**Answer**:
shared or single bathroom: 0.435
2 bathrooms: 0.476
3 or more bathrooms: 0.089

## C.2 AMERICAN TIME-USE SURVEY

**Split**: 2020

**Target**:
Labor Force Status : Employed; Unemployed; Not in Labor Force

**Conditions**:
Age = 30 - 44

**Natural Language Question**:
For a person aged 30-44 in the United States population in 2020, what is the probability distribution of their Labor Force Status?

**Answer**:
Employed: 0.797
Unemployed: 0.051
Not in Labor Force: 0.153

## C.3 WORLD VALUE SURVEY

**Split**: Sweden

**Target**:
Importance of politics in life : Not at all important; Not very important; Rather important; Very important

**Conditions**:
Member of humanitarian or charitable organization = Member

**Natural Language Question**:
For a person in Sweden aged 18 or older in 2010-2014 who is not a member of a humanitarian or charitable organization, what is the probability distribution of their views on the importance of politics in their life?

**Answer**:
Not at all important: 0.11
Not very important: 0.299
Rather important: 0.441
Very important: 0.15

