# OpenReview forum: "Let’s Think Var-by-Var: Large Language Models Enable Ad Hoc Probabilistic Reasoning"
_ICLR.cc/2025/Conference — Submitted to ICLR 2025_

### Official Review · Reviewer_XEun · 2024-10-27

**Soundness:** 3
**Presentation:** 4
**Contribution:** 2
**Rating:** 5
**Confidence:** 4

**Summary:**

This paper focuses on questions such as guesstimation questions (e.g. how much are airbnb listings in NJ?), where answering the question requires common sense reasoning where one needs to brainstorm additional useful variables. The paper proposes an ad hoc probabilistic model, where an LLM is used to brainstorm relevant variables, constrain the marginal (and conditional) distribution, and then build a probabilistic model from this to answer the question. This is in contrast to using the LLM to directly answer the question. The proposed method is evaluate on three real world datasets, where it mathes the performance of a chain-of-thought (CoT) based baseline.

**Strengths:**

1. The proposed method is an interesting way to combine the benefits of LLMs (which could be good at brainstorming relevant variables), and the benefits of probabilistic reasoning (where you have a systematic way to understand how the answer was obtained).

2. Once additional variables are proposed, the paper proposes to constrain the distribution rather than the parameters. This is interesting and makes sense because LLMs might be helpful in generating constraints for the distribution but not parameters.

3. The paper is very well written, and easy to follow, and Figure 1 does a great job of summarizing the proposed framework.

4. The paper performs useful ablations to understand if and how LLMs help in each part of the pipeline (sec 5.2).

**Weaknesses:**

1. The proposed method doesn’t perform better than a simple 0-shot CoT baseline. Additionally, the baseline considered here is not necessarily the strongest. For example, the baseline uses gpt-4o-mini, and I’m guessing a stronger model like gpt-4o or o1 might perform much better for guesstimation questions. The paper also uses a 0-shot baseline, and I imagine adding few-shot examples could also improve performance.

2. The datasets considered and the question generation pipeline might not be the best for the kind of questions this approach aims to solve. If I understand the question generation correctly (line 340), this generation process does not guarantee that the questions actually require brainstorming additional latent variables.

   a. Given that the focus of the work is on guesstimation questions, it’s also not clear why the paper doesn’t directly use a set of guesstimation questions available online (perhaps the proposed method helps more there!).

3. Using LLMs to brainstorm relevant variables and choosing quantiles to constrain are important components of the proposed method. Unfortunately the ablations in Figure 3 (column 1 and 2) where either the proposed latent variable or the direction of query to constraint are changed doesn’t seem to affect results too much.

4. Overall, the paper mentions lots of things which could have been explored and are left for future work (while this is not a weakness by itself, I think combined with the empirical results it does become a weakness.) E.g. f_i considered here are always indicator even though the framing is general enough to consider constraining mean/variance, the paper only explores using a small no. of categorical variables etc.

**Questions:**

1. Missing citations on line 253, and line 255.

2. Line 470 — why not an additional ablation where only the brainstormed queries are moved towards the oracle but not the query corresponding to Q?

3. Line 366 — how much does the averaging help / hurt performance?

4. Sec 5.2 writing suggestion — it might be helpful to name the stages. Right now the reader has to go back and forth to figure out what stage (a), (b) etc. correspond to.

5. Line 415 — could one reason why LLMs might be already answering these questions directly be that they seen the data in pretraining. E.g. ATUS, do you observe any difference on which year’s data was used? Is there a 2024 dataset available to test?

---

### Official Review · Reviewer_2Fvk · 2024-11-02

**Soundness:** 2
**Presentation:** 3
**Contribution:** 2
**Rating:** 5
**Confidence:** 4

**Summary:**

The paper proposes synthesizing an ad hoc probabilistic model with the assistance of LLMs to answer a guesstimation question, where LLMs are used to propose variables, construct directed graphs, and assign numerical conditional probabilities. Based on the LLM outputs, the joint distribution is optimized to maximize overall constraint satisfaction. The authors further demonstrate that their framework performs comparably to a direct CoT prompting baseline regarding TVD evaluation.

**Strengths:**

The method works on probabilistic inference with LLM, which is an important research topic with fewer current related works.

It’s innovative to obtain constraints from an LLM rather than from data samples, as LLMs contain extensive commonsense knowledge and can be viewed as a world model. While there are related works with similar ideas (https://arxiv.org/pdf/2402.02392, https://arxiv.org/pdf/2404.12494), these can be seen as concurrent works.

**Weaknesses:**

The primary weakness of the paper is that all experiments indicate the proposed method is not effective, particularly in the ablation studies.
- In Figure 3, columns 1 and 2 suggest that the final probability derivation is not strongly influenced by the variables or their relationships, which seems counterintuitive. This may imply that neither direct prompting nor the proposed method works effectively (see my question 1,2).
- Ablation 3 does not assess the accuracy of the elicited numeric constraints, as LLMs are known to be unreliable in directly estimating probabilities (https://arxiv.org/pdf/2306.13063).
- The explanation in lines 415–417 is somewhat unconvincing; the method still depends on the model’s ability to decompose the target question to identify the variables, implying that the model must have a sense of how to answer the target question.

**Questions:**

1. Could you explain in detail how you select the variables that are neither the target nor the condition in Ablation 1?

2. Could you provide an upper bound (worst-case scenario) for the TVD of your method across all three datasets, assuming that all variables are randomly selected and relationships in the variable graphs are randomly assigned?

3. Could you show some examples of where your method works better than the direct method?

4. Given that the results of Ablation 2 suggest that the directions of variables may not be beneficial, have you considered just assuming prior knowledge as conditionally independent on x?

---

### Official Review · Reviewer_3skv · 2024-11-05

**Soundness:** 2
**Presentation:** 3
**Contribution:** 2
**Rating:** 3
**Confidence:** 4

**Summary:**

The paper proposes a novel framework that leverages large language models (LLMs) for ad hoc probabilistic reasoning. The authors introduce a method to extract commonsense knowledge from LLMs and integrate it into probabilistic models to answer guesstimate questions. The approach involves prompting LLMs to propose relevant variables and constraints, which are then used to optimize a probabilistic model.

**Strengths:**

* The paper presents an innovative approach by using LLMs for direct question answering and as a source of commonsense knowledge for building probabilistic models.

**Weaknesses:**

**This work feels incomplete**

1. No analysis of why the proposed work did not work. The authors acknowledge it and plan to do it in the future.
2. The effectiveness of the framework heavily relies on the quality and capabilities of the LLM used. Variations in LLM performance could lead to inconsistent results, which is not addressed in the paper. Consider incorporating additional evaluation metrics or qualitative assessments to provide a more rounded view of model performance.
3. Why are LLMs instructed to propose at most four variables only? How did the authors come up with the number 4 and for each variable with at most 5 values?
4. Although the paper mentions the potential for handling continuous variables, the focus remains on discrete variables. A more detailed exploration or examples involving continuous data could strengthen the applicability of the method.
5. Errors can propagate from any stage (extracting relevant variables, choosing quantities to constrain) for the proposed method. How accurate are these stages, a deeper analysis is required.

**Questions:**

Typos:
1. Incorrect citations in lines 253 and 255. There are "?" instead of citations.

---

### Official Review · Reviewer_LB5G · 2024-11-08

**Soundness:** 1
**Presentation:** 2
**Contribution:** 2
**Rating:** 3
**Confidence:** 3

**Summary:**

The paper explores how large language models (LLMs) can support commonsense-based probabilistic reasoning. The authors propose a framework where LLMs generate ad hoc probabilistic models by identifying and defining relevant random variables and their interactions for specific guesstimation questions. The proposed method prompts the LLM to (1) propose a set of relevant variables and potential dependencies and (2) define constraints on the model's probability distribution using the variables. Then, the model is optimized to satisfy these constraints within a log-linear family.

The first experiment assesses the performance of the ad hoc probabilistic reasoning pipeline on three real-world datasets. The method's results are compared against a baseline method using zero-shot Chain-of-Thought (CoT) prompting. Results show that while the framework does not outperform direct prompting on simpler questions, it performs comparably without introducing significant errors.

The second set of experiments investigates the robustness of specific stages in the framework by introducing controlled modifications in how variables and constraints are selected. The results show that randomizing variables or flipping constraint directions does not drastically impact performance, suggesting that the framework is relatively robust to these changes.

**Strengths:**

1. The paper proposes a unique framework that leverages LLMs to create probabilistic models on the fly. This allows for a flexible approach to questions requiring commonsense reasoning.
2. The paper contributes to an important and popular topic on scaling inference-time computing and the trend of achieving slow reasoning over fast thinking. The methods and findings can potentially be of interest to communities looking for an optimization and refinement-focused reasoning method.

**Weaknesses:**

1. The main results are not strong enough. On the end-to-end evaluation with real-world data, the method does not outperform the direct CoT approach on average. A few improvements (delta TVD) seem pretty marginal. Thus, the proposed method's benefits are not conclusive. Similarly, the delta TVDs for the intervention analysis aren't significant enough to reach a clear and strong conclusion.
2. The proposed method of "slow-thinking" through a probabilistic model can potentially have a new impact on the direction of inference-time compute scaling and fast-to-slow reasoning. However, the paper's experiments do not highlight the impact well. The contribution to the field is not clear or relatively weak. Based on the experiments conducted, it's not clear how the proposed model can connect to a useful downstream application. I believe that experiments focusing on tasks that are more connected to downstream usability or improving the model's reasoning ability on complex problems would strengthen the paper's contributions.
3. Currently, it seems that the model requires a problem to have candidate answers (multiple choices). The experiments did not show how the proposed method performs in an unknown search space with an unknown number of candidates. It would make the method's robustness and scalability clearer if the authors could show how we can generalize from problems with clear candidates to ones with unclear answer options.
4. The choice of data seems suboptimal. The method utilizes commonsense knowledge within the model parameters. Yet, the questions evaluated are more factual and information-related categories. Such questions seem to fit better with a model that can utilize information from a search engine or RAG system. Meanwhile, some scenarios fit better with the motivation of depending on the model's internal knowledge and world model. For example, one situation could be using the model's physical understanding to infer which car is to blame in a traffic accident, given the dynamics and positions of the vehicles. Otherwise, the paper would benefit from strengthening the motivation of relying on the model's internal knowledge rather than its tool-using ability.

**Questions:**

1. Did you consider generating the random variables by providing additional context from a search engine or RAG database to the LLM?

---

### Author Response · Authors · 2024-12-04

We thank all the reviewers for the thoughtful and constructive feedback. Like you, we were disappointed that so far, our method has not shown meaningful benefits over the zero-shot Chain-of-Thought baseline.

We used public datasets to facilitate systematic evaluation, but as Reviewer XEun pointed out, LLMs may already know the factual information in these datasets well. We still think our method of constructing articulated ad hoc models may show benefit in scenarios that require reasoning over different cases, or, as Reviewer LB5G suggested, scenarios that demand intuitive physics and Theory-of-Mind capabilities. We plan to test the method on more datasets, including those used in BIRD, a concurrent and related work mentioned by Reviewer 2Fvk.

We appreciate Reviewer 3skv's observation regarding the restriction to 4 variables and 5 values in our experiments. Our pipeline involved many design choices as well, e.g., prompting strategies, graph sizes. It may help to tune those choices with hyperparameter searches. Similarly, we plan to experiment with the constrained optimization setup in order to take LLMs’ confidence measures into account and explore alternative loss functions.

---

### Meta-Review · Area_Chair_9wrQ · 2024-12-17

**Metareview:**

This paper proposes answering guesstimation questions by prompting a language model to come up with a probabilistic graphical model (variables, conditional and marginal distributions, and parameter optimization) and doing inference in it. The paper is well written and the idea is sound. However, the proposed method doesn’t outperform direct chain-of-thought prompting.

**Additional Comments On Reviewer Discussion:**

The reviewers find the paper interesting and well-motivated but agree that the results are not good enough. The reviewers also suggest follow up directions.

---

### Decision · Program_Chairs · 2025-01-22

Reject